# ZERO BUBBLE (ALMOST) PIPELINE PARALLELISM

**Penghui Qi**[*], **Xinyi Wan**[*], **Guangxing Huang & Min Lin**
Sea AI Lab
{qiph,wanxy,huanggx,linmin}@sea.com

## ABSTRACT

Pipeline parallelism is one of the key components for large-scale distributed training, yet its efficiency suffers from pipeline bubbles which were deemed inevitable. In this work, we introduce a scheduling strategy that, to our knowledge, is the first to successfully achieve zero pipeline bubbles under synchronous training semantics. The key idea behind this improvement is to split the backward computation into two parts, one that computes gradient for the input and another that computes for the parameters. Based on this idea, we handcraft novel pipeline schedules that significantly outperform the baseline methods. We further develop an algorithm that automatically finds an optimal schedule based on specific model configuration and memory limit. Additionally, to truly achieve zero bubble, we introduce a novel technique to bypass synchronizations during the optimizer step. Experimental evaluations show that our method outperforms the 1F1B schedule up to 15% in throughput under a similar memory limit. This number can be further pushed to 30% when the memory constraint is relaxed. We believe our results mark a major step forward in harnessing the true potential of pipeline parallelism. The source code based on Megatron-LM is publicly avaiable at https://github.com/sail-sg/zero-bubble-pipeline-parallelism.

## 1 INTRODUCTION

The realm of distributed model training has become a focal point in the deep learning community, especially with the advent of increasingly large and intricate models. Training these behemoths often requires a vast amount of GPUs interconnected with various topologies. Various parallelism techniques have been proposed for training DNN in the past years. Data parallelism (DP) (Goyal et al., 2017; Li et al., 2020) is the default strategy for models of small to moderate sizes due to its simplicity. Beyond a model size, it is no longer possible to fit the model parameters in one single GPU. This is when model parallelism comes to the rescue (Harlap et al., 2018; Huang et al., 2019; Fan et al., 2021; Zheng et al., 2022). There are two main model parallel schemes, tensor parallelism (TP) and pipeline parallelism (PP). TP splits the matrix multiplication in one layer to several devices, while PP segments the entire model into different stages which can be processed across different devices. Notably, ZeRO (Rajbhandari et al., 2020) provides a strong alternative to model parallelism by sharding parameters across devices, while keeping the simplicity of DP.

Recent research indicates that achieving optimal performance in large-scale training scenarios requires a non-trivial interaction of DP, TP and PP strategies. In the abundance of interconnection resources, e.g. NVLink between GPUs within one compute node, a hybrid of DP, TP and ZeRO strategies works efficiently. Whereas there are numerous empirical evidences Fan et al. (2021); Zheng et al. (2022); Narayanan et al. (2021) showing PP is particularly advantageous for utilizing cross-server connections, especially at the scale of thousands of GPUs. This highlights the primary aim of our work: enhancing the efficiency of PP.

Going deeper into the intricacies of PP, the efficiency of its implementation relies heavily on the amount of device idle time referred to as pipeline bubbles. Due to the dependency between layers, bubbles seem inevitable. A prominent early work to address this issue is GPipe (Huang et al., 2019), which attempts to reduce the bubble ratio by increasing the number of concurrent batches in the pipeline. However, a direct consequence of this is an increase in peak memory demands.

---

[*]Equal Contributors

To mitigate this, GPipe discards part of the intermediate activations while recomputing them during the backward pass. Yet, this approach introduced a computation overhead of around 20% (Fan et al., 2021). One line of work that improves over GPipe focuses on asynchronous PP, including PipeDream (Harlap et al., 2018), PipeMare (Yang et al., 2021). Asynchronous PP is theoretically bubble free, they greatly improve pipeline efficiency, however, at the sacrifice of exact optimization semantics. On the other hand, improvements are also made under synchronous settings. A notable scheduling strategy to address the limitation of GPipe is called *one-forward-one-backward (1F1B)*. It was first proposed in PipeDream (Harlap et al., 2018) under the asynchronous setting, and later introduced under synchronous settings (Fan et al., 2021; Narayanan et al., 2021). 1F1B offers faster memory clearance by early scheduling the backward passes. With the same number of microbatches, it yields similar bubble ratios but with a distinct advantage in peak memory. Based on 1F1B, Narayanan et al. (2021) introduced the 1F1B *interleaved* strategy. By assigning multiple stages to the same device, it further reduces the bubble size at the cost of more communication and higher peak memory.

Despite various efforts, to this date the remaining bubbles still pose the largest issue for PP under synchronous training semantics. In this work, we spotted the opportunity that PP can be further optimized by representing and scheduling the computation graph at a finer granularity. Classical deep learning frameworks are designed at the granularity of layers, whereas modern deep learning compilers use different intermediate representations for optimizations at various levels. (Chen et al., 2018; Roesch et al., 2018; Sabne, 2020; Tillet et al., 2019; Lattner et al., 2020). Although a finer granularity always means a larger space for searching, it is often impeded by the lack of optimization tools to navigate the space. Therefore, choosing a suitable granularity is crucial.

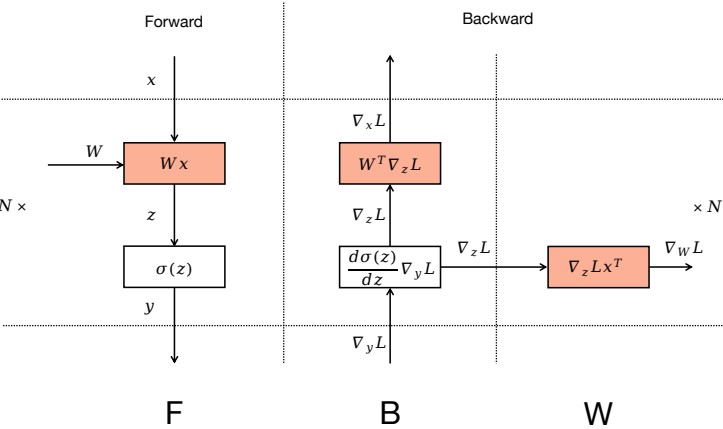

Figure 1: Computation Graph for MLP.

Traditionally, neural networks are granularized as stacked layers. There are two functions associated with each layer, forward and backward. In the forward pass, the input $x$ is transformed into the output $y$ with the parameterized mapping $f(x, W)$. The backward pass, crucial for training, involves two computations: $\nabla_x f(x, W)^\top \frac{d\ell}{dy}$ and $\nabla_W f(x, W)^\top \frac{d\ell}{dy}$. Correspondingly, they compute the gradient with respect to the input $x$ and the layer's parameters $W$. For convenience, we use single letters $B$ and $W$ to denote these two computations respectively, and $F$ to denote forward pass (Figure 1). Traditionally, $B$ and $W$ are grouped and provided as a single backward function. This design is conceptually friendly to the user, and it happens to work well for DP, because the communication of the weights' gradient at layer $i$ can be overlapped with the backward computation at layer $i - 1$. However, in PP, this design unnecessarily increases the sequentially dependent computations, i.e. $B$ at the layer $i - 1$ depends on $W$ at the layer $i$, which is usually detrimental for the efficiency of the pipeline.

Based on split $B$ and $W$, we present new pipeline schedules that greatly improve pipeline efficiency. The remainder of this paper is organized as follows: In Section 2, we introduce handcrafted schedules based on an ideal assumption that the execution times of $F$, $B$ and $W$ are identical. Subsequently,

in Section 3, we remove this assumption and propose an automatic scheduling algorithm that works under more realistic conditions. To achieve zero bubble, Section 4 details a method that sidesteps the need for synchronization during the optimizer step, yet preserves synchronous training semantics. We conclude with empirical evaluations of our methods against baseline methods under diverse settings.

We should note that we do not aim to explore general mixed strategies for large scale distributed training. Instead, we specifically target to improve the efficiency of pipeline scheduling, supported with apple to apple comparisons with baselines. Our method is orthogonal to DP, TP and ZeRO strategies, and it can be used as a parallel replacement for the PP part in large scale training.

## 2 HANDCRAFTED PIPELINE SCHEDULES

Based on the key observation that splitting $B$ and $W$ could reduce sequential dependency and thus improve efficiency, we redesign the pipeline starting from the commonly utilized 1F1B schedule. As depicted in Figure 2, 1F1B initiates with a warm-up phase. In this phase, workers conduct varying numbers of forward passes, with each stage typically performing one more forward pass than its immediately subsequent stage. Following the warm-up phase, each worker transits to a steady state where they alternately execute one forward pass and one backward pass, ensuring an even workload distribution among stages. In the final phase, each worker processes the backward passes for the outstanding in-flight microbatches, completing the batch.

In our improved version we split the backward pass into $B$ and $W$ passes, it is imperative that $F$ and $B$ from the same microbatch must still remain sequentially dependent across pipeline stages. However, $W$ can be flexibly scheduled anywhere after the corresponding $B$ of the same stage. This allows for strategic placement of $W$ to fill the pipeline bubbles. There are many possible schedules that improve over 1F1B, trading off differently on the bubble size and the memory footprint. We introduce two particularly interesting handcrafted schedules in this section to show the great potential of finer granularity at reducing pipeline bubbles (see Figure 3). For the sake of clarity in our initial design, we assume that the time costs for $F$, $B$, and $W$ are identical, an assumption shared by earlier studies (Narayanan et al., 2021; Huang et al., 2019). However, in Section 3, we re-evaluate this assumption to optimize scheduling efficiency in real-world scenarios.

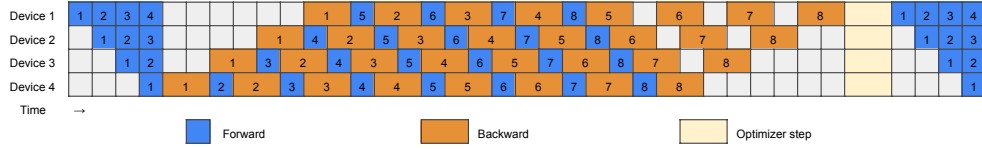

Figure 2: 1F1B pipeline schedule.

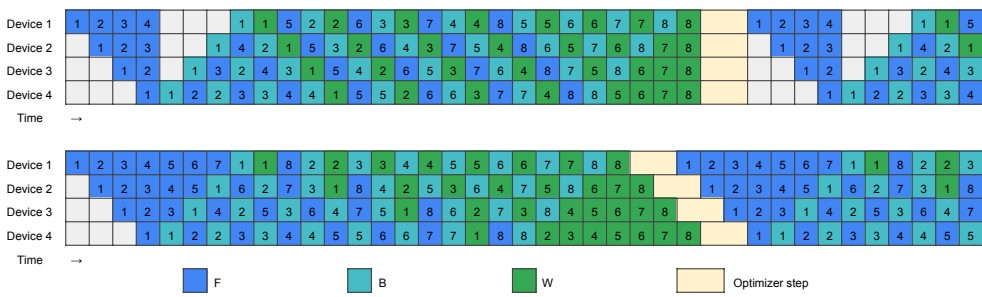

Figure 3: Handcrafted pipeline schedules, top: ZB-H1; bottom: ZB-H2

### 2.1 MEMORY EFFICIENT SCHEDULE

Our first handcrafted schedule, named ZB-H1, ensures that the maximum peak memory usage over all workers doesn't exceed that of 1F1B. ZB-H1 generally follows the 1F1B schedule, but it adjusts

the starting points of *W* depending on the number of warm-up microbatches. This ensures all workers maintain the same number of in-flight microbatches. As a result, as seen in Figure 3 (top), the bubble size is reduced to a third of 1F1B's size. This reduction is because *B* is initiated earlier across all workers compared to 1F1B, and the tail-end bubbles are filled by the later-starting *W* passes. As *W* typically uses less memory than *B* (Table 1), the first worker has the maximum peak memory usage which is consistent with 1F1B.

## 2.2 ZERO BUBBLE SCHEDULE

When we permit a larger memory footprint than 1F1B and have a sufficient number of microbatches, it's possible to achieve a zero bubble schedule, which we label as ZB-H2. As illustrated in Figure 3 (bottom), we introduce more *F* passes during the warm-up phase to fill the bubble preceding the initial *B*. We also reorder the *W* passes at the tail, which changes the layout from trapezoid into a parallelogram, eliminating all the bubbles in the pipeline. It is important to highlight that the synchronization between the optimizer steps is removed here, we discuss how this is safely done in Section 4.

## 2.3 QUANTITATIVE ANALYSES

We use $p$ to denote the number of stages and $b$ to denote the size of each microbatch. For transformer architecture, we denote the number of attention heads as $a$, the sequence length as $s$ and the hidden dimension size as $h$. We use the notations $M_B/M_W$ to represent the memory required to store activations for one *B/W* pass, and $T_F/T_B/T_W$ to represent the running time for one *F/B/W* pass. For simplicity, we only do quantitative analyses on transformer architecture (Vaswani et al., 2017), using a typical setting similar to GPT-3 (Brown et al., 2020) where the hidden dimension size inside feedforward is $4h$ and the dimension size for each attention head is $h/a$.

As in Narayanan et al. (2021), we only consider matmul operations when calculating FLOPs because they contribute most of the computations in a transformer layer. For each matmul operation in the forward pass, there are two matmul operations with the same FLOPs in corresponding backward pass (see Figure 1), each of which belongs to either *B* or *W*. The approximate formula for calculating the FLOPs of a transformer layer is in Table 1. We can see that $T_W < T_F < T_B$ and $T_B + T_W = 2T_F$. We use the same method in Korthikanti et al. (2023) to estimate activations memory required for *B*. After *B* completes, it releases some activations not used anymore but keeps some extra gradients ($\nabla_z L$ in Figure 1) for *W*. The total memory required by *W*, as in Table 1, is less than *B*.

Table 1: FLOPs and activations memory required per transformer layer for each pass

| Pass | FLOPs | Activations Memory Required |
|:---:|:---:|:---:|
| *F* | $sbh(24h + 4s)$ | 0 |
| *B* | $sbh(24h + 8s)$ | $sb(34h + 5as)$ |
| *W* | $sbh(24h)$ | $32sbh$ |

Without the assumption of $T_F = T_B = T_W$, the peak activations memory and bubble size of ZB-H1 and ZB-H2 are quantified in Table 2. Notably, the activations memory of worker $i$ is $(p - i + 1)M_B + (i - 1)M_W$ for ZB-H1 and $(2p - 2i + 1)M_B + (2i - 2)M_W$ for ZB-H2. As in Table 1, the activations memory required for *W* is smaller than that for *B*. Therefore, the peak activations memory is $pM_B$ and $(2p - 1)M_B$, for ZB-H1 and ZB-H2 respectively.

Table 2: Comparison between 1F1B and our handcrafted schedules.

| Schedule | Bubble size | Peak activations memory |
|:---:|:---:|:---:|
| 1F1B | $(p-1)(T_F + T_B + T_W)$ | $pM_B$ |
| ZB-H1 | $(p-1)(T_F + T_B - T_W)$ | $pM_B$ |
| ZB-H2 | $(p-1)(T_F + T_B - 2T_W)$ | $(2p-1)M_B$ |

## 3 AUTOMATIC PIPELINE SCHEDULING

While handcrafted schedules offer simplicity and better comprehensibility, they face several issues in practical applications. For one, scheduling under the assumption that $T_F = T_B = T_W$ introduces unwanted bubbles, especially for models where these values differ significantly. Moreover, communication time (denoted as $T_{comm}$) required to transfer activation/gradient between stages is often ignored in handcrafted schedules, leading to noticeable latencies in the pipeline stream. Finally, striking a balance between minimizing bubble size and adhering to memory limit becomes particularly challenging when the available memory is insufficient to accommodate enough microbatches for a bubble-free schedule.

To address these challenges and ensure generalization to practical scenarios, we propose algorithms to automatically search the optimal schedule given the number of pipeline stages $p$, the number of microbatches $m$, the activations memory limit $M_{limit}$, and the running time estimations $T_F$, $T_B$, $T_W$ and $T_{comm}$. We design a heuristic strategy, which always generates an optimal or near optimal solution especially when $m$ is large enough. We also systematically formulate the problem as Integer Linear Programming (for more details see Appendix G), which can be solved by an off-the-shelf ILP solver (Forrest & Lougee-Heimer, 2005) when the problem is under a certain scale. These two approaches can be combined: first, use the heuristic solution as initialization, and then optimize it further with ILP.

### 3.1 THE HEURISTIC ALGORITHM

We present our heuristic algorithm in the following steps:

- In the warm-up phase, within the memory limit, we schedule as many $F$ passes as possible to minimize the bubble before the first $B$. The resulting schedule may still have a small bubble (less than $T_F$) before the first $B$ if not reaching memory limit, where scheduling another $F$ may delay the following $B$. We use a binary hyperparameter to control whether to do it or not.

- After the warm-up phase, we adhere to the pattern where one $F$ and one $B$ are scheduled iteratively. We insert $W$ to fill the bubble when there is a gap larger than $T_W$. When a bubble occurs but the size is less than $T_W$, we still insert a $W$ if the current bubble makes the largest cumulative bubble size among all stages become larger. We also insert $W$ to recycle some memory when the memory limit is hit. Typically, our heuristic strategy enters a steady state that follows $1F$-$1B$-$1W$ pattern.

- Throughout this process, pipeline stage $i$ is always guaranteed to schedule at least one more $F$ than stage $i + 1$ anytime before $F$ is used up. When this difference exceeds one, we use another binary hyperparameter to decide whether to skip one $F$ in pipeline stage $i$ if it doesn't cause more bubbles. We perform a grid search to find the best combination of hyperparameters.

- In each stage, when $F$ and $B$ passes run out, we schedule all the left $W$ passes one by one.

## 4 BYPASSING OPTIMIZER SYNCHRONIZATIONS

In most practices of PP, synchronizations over pipeline stages are usually performed in optimizer step for the sake of numerical robustness. For example, a global gradient norm needs to be computed for gradient norm clipping (Pascanu et al., 2013); a global check for NAN and INF values are performed in the mixed precision settings (Micikevicius et al., 2017); both of them require an all-reduce communication across all stages. However, synchronization at the optimizer step destroys the parallelogram (Figure 3) and makes zero bubble impossible. In this section, we propose an alternative mechanism to bypass these synchronizations, while still maintaining a synchronous optimization semantics.

In existing implementations, an all-reduce communication is first launched to collect the global states, followed by the optimizer steps which are conditioned on the global states. However, we noticed that most of the time the global states have no effects, e.g., the global check for NAN and INF rarely trigger because in a robust setting most iterations shouldn't have numerical issues; the gradient clipping rate is also quite low empirically to justify a synchronization of global gradient norm at every iteration.

Based on these observations, we propose to replace the before-hand synchronizations with a post update validation. The idea is illustrated in Figure 4, at each stage before the optimizer step, a partially reduced global state is received from the previous stage, combined with the current stage's local state, and passed on to the next stage. The optimizer step of each stage is controlled by the partially reduced state, e.g. skip the update when a NAN is spotted or the partially reduced gradient norm exceeds the clipping threshold. During the warm-up phase of the next iteration, the fully reduced global state is then propagated back from the last stage to first stage. Upon receiving the global state, each stage performs a validation to decide whether the previous optimizer step is legitimate. If an amendment to the gradient is required, a rollback will be issued (for more details see Appendix C) and then we redo the optimizer step based on the fully reduced global state.

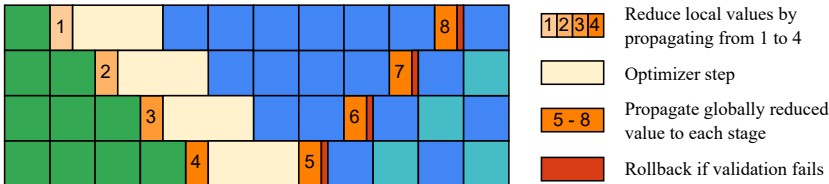

Figure 4: The post-validation strategy to replace optimizer synchronization.

## 5 EXPERIMENTS

### 5.1 SETUP

We base our implementation on the open-source Megatron-LM project (Narayanan et al., 2021) and assess its performance using models analogous to GPT-3 (Brown et al., 2020), as detailed in Table 3. During our experiments, we first conducted a specific number of iterations for profiling, collecting empirical measurements for $T_F$, $T_B$, $T_W$, and $T_{comm}$. After obtaining these values, we fed them into our automatic pipeline scheduling algorithm to determine the optimal schedule. It's worth noting that both the initial and final pipeline stages possess one fewer transformer layer compared to the intermediate stages. This design is to compensate for the extra embedding lookup and loss computations in the initial and final stages so that they won't become the bottleneck and cause bubbles to other stages.

Table 3: Models and fixed settings used in experiments

| Model | Layers | Attention Heads | Hidden Size | Sequence Length | Pipelines (GPUs) | Microbatch Size | Number of Microbatches |
|-------|--------|-----------------|-------------|-----------------|------------------|-----------------|------------------------|
| 1.5B  | 22     | 24              | 2304        | 1024            | 8                | 6               | 24 / 32 / 64           |
| 6.2B  | 30     | 32              | 4096        | 1024            | 8                | 3               | 24 / 32 / 64           |
| 14.6B | 46     | 40              | 5120        | 1024            | 16               | 1               | 48 / 64 / 128          |
| 28.3B | 62     | 48              | 6144        | 1024            | 32               | 1               | 96 / 128 / 256         |

Compared methods:

- ZB-1p: Automatically searched schedule with the activation memory limited to $pM_B$, which theoretically has the same peak memory as 1F1B.

- ZB-2p: Automatically searched schedule with the activation memory limited to $2pM_B$, which is the least amount of memory to empirically achieve close to zero bubble (see Figure 7).

- 1F1B and 1F1B-I: 1F1B and interleaved 1F1B methods introduced by Harlap et al. (2018) and Narayanan et al. (2021) with implementation from Megatron-LM. For interleaved 1F1B, the entire model is divided into a sequence of chunks, which are cyclically taken by each stage, forming an interleaved pipeline. In our interleaved experiments, we always use the maximum number of chunks to ensure least bubble, i.e. each transformer layer serves as a chunk.

Our experiments utilize up to 32 NVIDIA A100 SXM 80G GPUs distributed across 4 nodes interconnected by a RoCE RDMA network. The running time of each iteration is recorded after several warm-up iterations. Thanks to the reproducibility provided by Megatron-LM implementation, we can verify the correctness of ZB-1p and ZB-2p without running models until convergence. We use a fixed random seed to initialize the model, record the loss after every iteration for ZB-1p, ZB-2p, and 1F1B, and then verify that they're bit-to-bit identical.

## 5.2 MAIN RESULTS

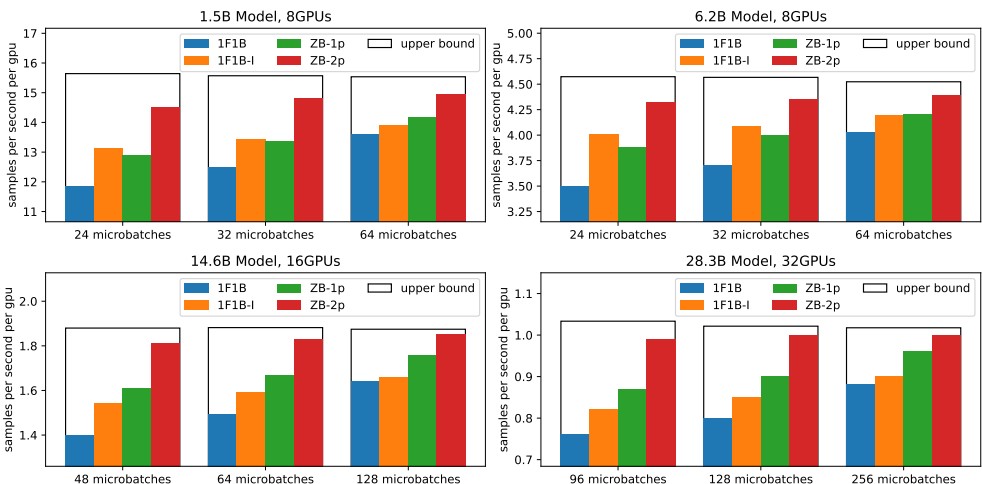

Figure 5: Comparison of throughput across different pipeline schedules.

Table 4: Experiment result details

| Setup | Model | 1.5B | | | 6.2B | | | 14.6B | | | 28.3B | | |
|---|---|---|---|---|---|---|---|---|---|---|---|---|---|
| | #GPU | 8 | | | 8 | | | 16 | | | 32 | | |
| | #Microbatch | 24 | 32 | 64 | 24 | 32 | 64 | 48 | 64 | 128 | 96 | 128 | 256 |
| Samples per GPU per second | ZB-2p | **14.5** | **14.8** | **14.9** | **4.32** | **4.35** | **4.39** | **1.81** | **1.83** | **1.85** | **0.99** | **1.00** | **1.00** |
| | ZB-1p | 12.9 | 13.4 | 14.2 | 3.88 | 4.00 | 4.20 | 1.61 | 1.67 | 1.76 | 0.87 | 0.90 | 0.96 |
| | 1F1B | 11.8 | 12.5 | 13.6 | 3.50 | 3.70 | 4.03 | 1.40 | 1.49 | 1.64 | 0.76 | 0.80 | 0.88 |
| | 1F1B-I | 13.1 | 13.4 | 13.9 | 4.01 | 4.08 | 4.19 | 1.54 | 1.59 | 1.66 | 0.82 | 0.85 | 0.90 |
| Memory (GB) | ZB-2p | 59 | 59 | 59 | 70 | 70 | 70 | 51 | 51 | 51 | 74 | 74 | 74 |
| | ZB-1p | 32 | 32 | 32 | 42 | 42 | 42 | 33 | 33 | 33 | 44 | 44 | 44 |
| | 1F1B | **30** | **30** | **30** | **39** | **39** | **39** | **32** | **32** | **32** | **43** | **43** | **43** |
| | 1F1B-I | 40 | 40 | 40 | 48 | 48 | 48 | 39 | 39 | 39 | 58 | 58 | 58 |

We present the throughput of all methods in Figure 5, and leave the additional details for each setup in Table 4. Our experiments demonstrate that ZB-2p consistently outperforms all other methods across various settings. Notably, the throughput of 1F1B, 1F1B-I and ZB-1p show a strong positive correlation with the number of microbatches. In contrast, ZB-2p maintains the efficiency even with fewer microbatches. This is because the bubble rate in ZB-2p has almost reached zero (Table 5), and its throughput is already close to the upper bound. Here the upper bound is roughly estimated by multiplying the throughput of 1F1B and $\frac{1}{1-\text{bubble rate of 1F1B}}$ (for more details see Section 5.3). As mentioned before, the improved efficiency of ZB-2p comes at the cost of a higher memory consumption compared to the 1F1B baseline. We also compare ZB-2p with 1F1B under the same memory consumption in Appendix F, and the experimental results also show that ZB-2p achieves a higher throughput even with half microbatch size compared to 1F1B.

In contrast, ZB-1p is designed to have a peak memory cost similar to the 1F1B baseline. It shows a comparable throughput to 1F1B-I in the 8 GPUs setups. In multi-node setups where communication bandwidth is more of a bottleneck, ZB-1p clearly outperforms 1F1B-I, highlighting its advantage in reducing pipeline bubbles without incurring extra communication cost.

In most of our settings we set number of microbatches $m$ larger than number of stages $p$ because they're more common use cases of pipeline parallelism. However we conducted experiments listed in Appendix H for $m \leq p$ cases which shows 20% to 30% improvements with a similar memory consumption.

## 5.3  EFFICIENCY OF AUTOMATIC SCHEDULING

Table 5: Bubble rates of 1F1B, 1F1B-I, ZB-H1, ZB-H2, ZB-1p, ZB-2p under different settings.

| Model | #Stage ($p$) | #Microbatch ($m$) | 1F1B | 1F1B-I | ZB-H1 | ZB-H2 | ZB-1p | ZB-2p |
|---|---|---|---|---|---|---|---|---|
| 1.5B | 8 | 24 | 0.2431 | 0.1055 | 0.1585 | 0.1083 | 0.1585 | **0.0433** |
| | | 32 | 0.1985 | 0.0818 | 0.1242 | 0.0837 | 0.1242 | **0.0039** |
| | | 64 | 0.1240 | 0.0443 | 0.0674 | 0.0444 | 0.0674 | **0.0026** |
| 6.2B | 8 | 24 | 0.2347 | 0.0808 | 0.1323 | 0.0698 | 0.1323 | **0.0029** |
| | | 32 | 0.1898 | 0.0628 | 0.1045 | 0.0559 | 0.1045 | **0.0022** |
| | | 64 | 0.1091 | 0.0320 | 0.0554 | 0.0294 | 0.0554 | **0.0010** |
| 14.6B | 16 | 48 | 0.2552 | 0.1104 | 0.1397 | 0.0672 | 0.1397 | **0.0066** |
| | | 64 | 0.2082 | 0.0852 | 0.1088 | 0.0516 | 0.1088 | **0.0054** |
| | | 128 | 0.1251 | 0.0445 | 0.0576 | 0.0266 | 0.0576 | **0.0028** |
| 28.3B | 32 | 96 | 0.2646 | 0.1493 | 0.1421 | 0.0641 | 0.1421 | **0.0038** |
| | | 128 | 0.2168 | 0.1164 | 0.1106 | 0.0490 | 0.1106 | **0.0029** |
| | | 256 | 0.1352 | 0.0624 | 0.0594 | 0.0257 | 0.0594 | **0.0018** |

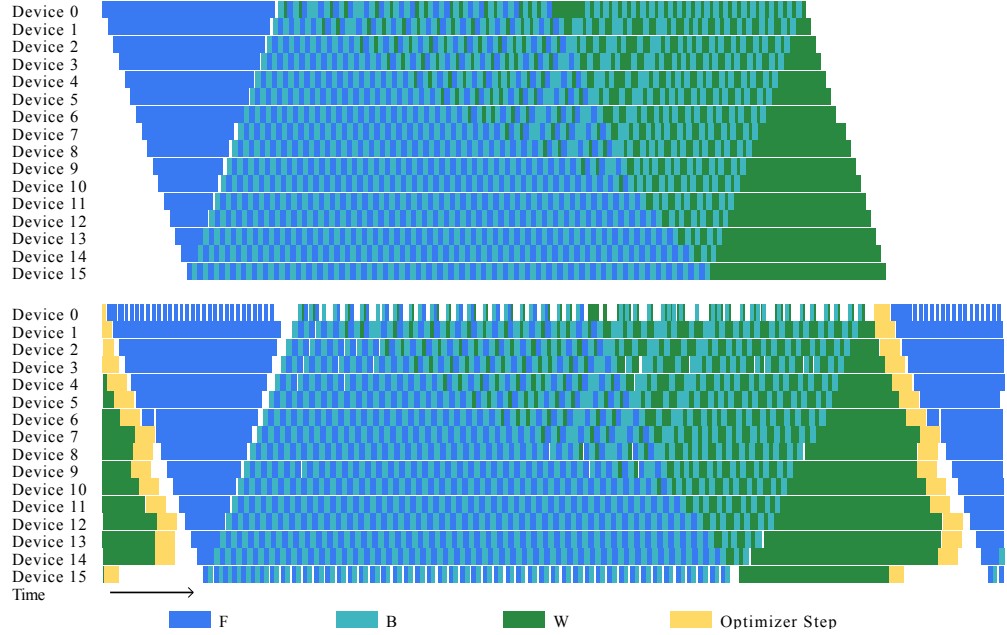

Figure 6: A schedule produced by ZB-2p (top) and its profiled execution process (bottom).

We study the efficiency of the schedules generated from our automatic scheduling algorithm. The same setups as our main experiments are used, however, since our purpose is to study the efficiency of the automatic scheduling algorithm, the numbers here are based on theoretical calculations instead of real experiments. To quantify the efficiency of a pipeline schedule, we introduce the concept of bubble rate, which is calculated as $(\text{cost} - m(T_F + T_B + T_W))/\text{cost}$. The cost here is defined as

the largest execution time of all stages, calculated for each schedule using profiled $T_F$, $T_B$, $T_W$ and $T_{\text{comm}}$ values. The $m(T_F + T_B + T_W)$ is the optimal execution time when all communications are overlapped with computations and hence no bubbles in the pipeline.

The bubble rates for different schedules are presented in Table 5. We include the handcrafted schedules ZB-H1 and ZB-H2 as baselines to the automatically searched schedules. In most of the settings, ZB-2p produces a bubble rate of less than 1%, which is the best among all schedules. In contrast, ZB-H2 consistently performs worse than ZB-2p. This provides a strong evidence that our automatic scheduling algorithm adapts better to realistic scenarios by using more accurate estimates of $T_F$, $T_B$, $T_W$ and $T_{\text{comm}}$. On the contrary, this improvement is not observed in ZB-1p vs ZB-H1, hypothetically because the memory limit becomes the dominate factor. Notably, all of our methods significantly outperform 1F1B.

We also plot ZB-2p and its profiled real execution on 16 GPUs to provide a direct visual evidence that it is truly a zero bubble schedule. As shown in Figure 6, the automatically generated ZB-2p schedule has almost no bubble. The profiled execution has slightly more bubbles but retains a good overall alignment.

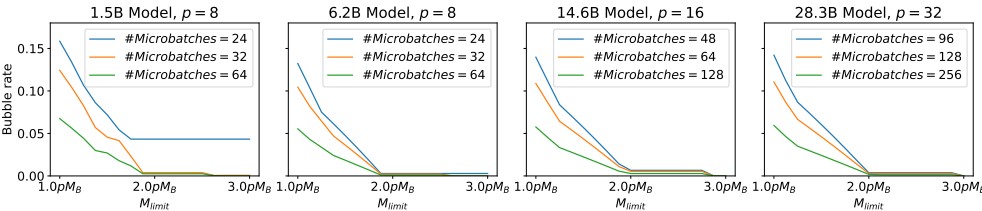

Figure 7: The relation between memory limit and bubble rate using our heuristic algorithm.

## 5.4 MEMORY LIMIT

To better understand the effect of memory limit, we study the relationship of the bubble rate to $M_{\text{limit}}$. We run our heuristic algorithm with a series of $M_{\text{limit}}$ and plot them in Figure 7. Initially, the bubble rate shows a close-to-linear decreasing trend as we increase the value of $M_{\text{limit}}$. Theoretically, the curve should plateau around $\frac{(p-1)(T_B + 2T_{\text{comm}}) + pT_F}{T_F} M_B$. Empirically, we find $2pM_B$ a good threshold for achieving close to zero bubble rate when $T_F \approx T_B$ and $T_{\text{comm}}$ is relatively small. Beyond the inflection point, although a sufficiently large memory limit does result in a theoretically zero bubble rate, in general the cost outweighs the gain. For more details see Appendix B.

## 6 CONCLUSION AND DISCUSSION

In this work, we introduced a novel strategy to improve the efficiency of pipeline parallelism by splitting the activation gradient and parameter gradient in backward computation, and we design an automatic pipeline scheduling algorithm that can minimize the pipeline bubble rate under different memory budgets. The schedules produced by this algorithm consistently outperform 1F1B and even achieve close to zero bubble rate. Empirically, achieving zero bubble requires approximately twice the activation memory compared to 1F1B, which raises concerns about out of memory issues. According to Appendix F, we believe it is worth trading some memory for a zero bubble pipeline schedule in the training of large models. Strategies like ZeRO, tensor parallelism can be used to accommodate the increased memory need. Another advantage of zero bubble schedule is that it can achieve optimal efficiency with a smaller number of microbatches (typically $3p$ is enough), which means more microbatches can be partitioned over data parallelism dimension. This brings a better scalability for the training of large models.

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

## A    OVERLAP COMMUNICATION IN DATA PARALLELISM

When data parallelism is taken into consideration, an all-reduce communication is launched to collect gradients before optimizer step. Generally, such communication is poorly overlapped with computation pass, resulting in a latency especially when the communication bandwidth is limited. As in Figure 3, usually a number of $W$ passes are scheduled at the tail of an iteration. For each $W$ pass, it consists of several independent computations calculating gradients for different parameters. As in Figure 8, We can reorder all of these computations to cluster those calculating the gradients for the same parameter, thus achieving the optimal overlapping between computation and communication.

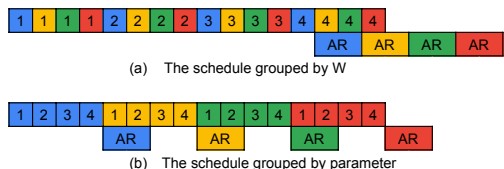

(a)    The schedule grouped by W

(b)    The schedule grouped by parameter

Figure 8: Comparison between the original schedule grouped by $W$ with poor overlapping (top) and the reordered schedule grouped by parameters with optimal overlapping (bottom). The number $i$ represents the computation belongs to $i$-th $W$, and different colors represent computations for different paramters.

## B    THE MEMORY LIMIT FOR AUTOMATIC SCHEDULING ALGORITHM

The relation between memory limit and bubble rate is highly affected by the bubbles preceding the first $B$ in the initial stage. For the first microbatch, the forward pass needs to go through from the initial stage to final stage, and the backward pass reverses this process until it eventually goes back to the initial stage. The total time for the first microbatch from start to complete takes at least $p(T_F + T_B) + 2(p-1)T_{\text{comm}}$ and it can not be squeezed due to the dependency chains. We denote the number of $F$ passes as $k(\geq 1)$ and the bubble size as $\beta(\geq 0)$, preceding the first $B$ pass in the initial stage. Then we have:

$$M_{\text{limit}} \geq kM_B \qquad (1)$$

$$\beta \geq p(T_F + T_B) + 2(p-1)T_{\text{comm}} - kT_F - T_B = (p-1)(T_B + 2T_{\text{comm}}) + (p-k)T_F \qquad (2)$$

The lower bound of $M_{\text{limit}}$ is in proportion to $k$ (see Formula 1), and $\beta$ is inversely proportional to $k$ (see Formula 2). When increasing $k$ and keeping $k < \lfloor \frac{(p-1)(T_B+2T_{\text{comm}})+pT_F}{T_F} \rfloor$, $\beta$ decreases linearly, meanwhile the lower bound of $M_{\text{limit}}$ increases linearly. When $k = \lfloor \frac{(p-1)(T_B+2T_{\text{comm}})+pT_F}{T_F} \rfloor$, $\beta$ reaches its minimum value without delaying $B$ and its value is less than $T_F$, with a peak activation memory at least $\lfloor \frac{(p-1)(T_B+2T_{\text{comm}})+pT_F}{T_F} \rfloor M_B$. Beyond this point, further reducing pipeline bubbles to zero is not easy. This is because there is a small bubble less than $T_F$ in each stage (see Figure 6), and scheduling another $F$ will delay the starting time of $B$ thus causing more requirements on $F$ in previous stages. Theoretically, another $p - 1$ F passes are required in the initial stage to fully eliminate bubbles preceding the first $B$ for all stages (see Figure 9), which also means a total activation memory usage at least $\lfloor \frac{(p-1)(T_B+2T_{\text{comm}})+(2p-1)T_F}{T_F} \rfloor M_B$.

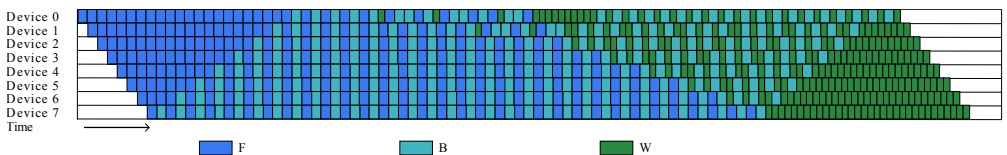

Figure 9: Zero bubble schedule for 1.5B model with 32 microbatches.

## C  IN-PLACE OPTIMIZER ROLLBACK

When we need to rollback an optimizer step, a typical method is to store a historic version of parameters and optimizer states, and revert to this historic version when needed. However, this method is memory inefficient and lots of copy operations are needed, which definitely hurts the training performance. For most optimizers, we notice that the step function is arithmetically reversible. Under this observation, we propose a novel technique to perform in-place optimizer rollback, which avoids allocating extra memory and requires extra computations only when the rollback is performed. As in Algorithm 1, we show how to rollback the step function for AdamW optimizer (Loshchilov & Hutter, 2017).

---

**Algorithm 1** In-place rollback for AdamW

1: **Optimizer States:**
2:     $\gamma(\text{lr})$, $\beta_1, \beta_2(\text{betas})$, $\epsilon$ (epsilon), $\lambda(\text{weight decay})$,
3:     $m$ (first moment), $v$ ( second moment), $\theta$ (parameters),
4:     $t$(time stamp).
5: **function** STEP$(g)$                                                    ▷ In-place step
6:     $t = t + 1$
7:     $m = \beta_1 m + (1 - \beta_1)g$
8:     $v = \beta_2 v + (1 - \beta_2)g^2$
9:     $m' = m/(1 - \beta_1^t)$
10:     $v' = v/(1 - \beta_2^t)$
11:     $\theta = \theta - \gamma\lambda\theta - \gamma m'/(\sqrt{v'} + \epsilon)$
12: **end function**
13: **function** ROLLBACK$(g)$                                              ▷ In-place rollback
14:     $m' = m/(1 - \beta_1^t)$
15:     $v' = v/(1 - \beta_2^t)$
16:     $\theta = (\theta + \gamma m'/(\sqrt{v'} + \epsilon))/(1 - \gamma\lambda)$
17:     $m = (m - (1 - \beta_1)g)/\beta_1$
18:     $v = (v - (1 - \beta_2)g^2)/\beta_2$
19:     $t = t - 1$
20: **end function**

---

## D  PROFILED TIME IN EXPERIMENTS

In our experiments, we record the profiled time of $T_F, T_B, T_W$, and $T_{\text{comm}}$ in ZB-2p across different settings. These values are then used to calculate bubble rates for all the methods considered in Section 5.3 and 5.4. These values can be found in Table 6.

Table 6: Profiled time of $T_F, T_B, T_W$, and $T_{\text{comm}}$.

| Model | #Stage ($p$) | #Microbatch ($m$) | $T_F$ | $T_B$ | $T_W$ | $T_{\text{comm}}$ |
|---|---|---|---|---|---|---|
| 1.5B | 8 | 24 | 18.522 | 18.086 | 9.337 | 0.601 |
|  |  | 32 | 18.513 | 18.086 | 9.331 | 0.626 |
|  |  | 64 | 18.546 | 18.097 | 9.321 | 0.762 |
| 6.2B | 8 | 24 | 29.718 | 29.444 | 19.927 | 0.527 |
|  |  | 32 | 29.802 | 29.428 | 19.530 | 0.577 |
|  |  | 64 | 29.935 | 29.621 | 19.388 | 0.535 |
| 14.6B | 16 | 48 | 11.347 | 11.248 | 8.132 | 0.377 |
|  |  | 64 | 11.307 | 11.254 | 8.101 | 0.379 |
|  |  | 128 | 11.325 | 11.308 | 8.109 | 0.378 |
| 28.3B | 32 | 96 | 10.419 | 10.207 | 7.715 | 0.408 |
|  |  | 128 | 10.408 | 10.204 | 7.703 | 0.408 |
|  |  | 256 | 10.402 | 10.248 | 7.698 | 0.460 |

# E   ABLATION STUDY ON OPTIMIZER POST-VALIDATION STRATEGY

In this section, we provide an ablation study on the effectiveness of the optimizer post-validation strategy. The study compares the throughput of ZB-2p under two conditions: with post-validation and with all-reduce synchronization. According to the experimental results in Table 7, the synchronized version of ZB-2p demonstrates a performance decrease of approximately 8% compared to ZB-2p with optimizer post-validation.

Table 7: Throughput (Samples per GPU per second) comparison between ZB-2p and synchronized ZB-2p

| Model | #Stage ($p$) | #Microbatch ($m$) | Post-validation | All-reduce synchronization |
|-------|--------------|-------------------|-----------------|----------------------------|
| 1.5B  | 8            | 24                | **14.5**        | 13.11                      |
| 6.2B  | 8            | 24                | **4.32**        | 4.00                       |
| 14.6B | 16           | 48                | **1.81**        | 1.68                       |
| 28.3B | 32           | 96                | **0.99**        | 0.91                       |

# F   COMPARE ZB-2P WITH 1F1B UNDER THE SAME MEMORY CONSUMPTION

Under the same memory consumption, we double the size of each microbatch for 1F1B and ZB-1p and compare their throughput with ZB-2p in Table 8. The experimental results show that ZB-2p also holds a better performance even with a half microbatch size compared to 1F1B. Empirically, a larger batch size increases the utilization rate of GPU and thus improves the efficiency. However, it is less of a concern for large models because the hidden dimension is large enough to saturate device utilization. Based on this consideration and our experimental results, we believe ZB-2p is more preferred than increasing the batch size for 1F1B. In some experiments where the device utilization is less saturated and $m/p$ is relatively large, ZB-1p with a doubled microbatch size may slightly outperform than ZB-2p.

# G   ILP FORMULATION

Any pass in the pipeline can be uniquely indexed by $(i, j, c)$, where $i \in \{1, 2, ..., p\}$ indexes the stage, $j \in \{1, 2, ..., m\}$ indexes the microbatch, and $c \in \{F, B, W\}$ denotes the specific pass of the microbatch. We define the variable $T_{(i,j,c)}$ as the time cost and $E_{(i,j,c)}$ as the ending time of a pass. We introduce $\Delta M_{(i,j,c)}$ to denote the memory increment incurred by the pass $(i, j, c)$. For example, $\Delta M_{(\cdot,\cdot,F)} = M_B$ because the forward pass leads to a net increase of $M_B$ of activation stored for the backward pass. $\Delta M_{(\cdot,\cdot,B)} = M_W - M_B$ which removes the memory stored for $B$ while adding those required by $W$, and $\Delta M_{(\cdot,\cdot,W)} = -M_W$. Finally, the variable that we want to search is the ordering of the passes in the schedule, for which we introduce the variable $O_{(i,j,c)\to(i,j',c')} \in \{0, 1\}$, which is an indicator whether the pass index by $(i, j, c)$ is scheduled before $(i, j', c')$.

$$\min_{O,E} \quad \max_i E_{(i,m,W)} - E_{(i,1,F)} + T_{(i,1,F)} \tag{3}$$

$$s.t. \quad E_{(i,j,F)} \geq E_{(i-1,j,F)} + T_{\text{comm}} + T_{(i,j,F)} \tag{4}$$

$$E_{(i,j,B)} \geq E_{(i+1,j,B)} + T_{\text{comm}} + T_{(i,j,B)} \tag{5}$$

$$E_{(i,j,c)} \geq E_{(i,j',c')} + T_{(i,j,c)} - O_{(i,j,c)\to(i,j',c')}\infty \tag{6}$$

$$M_{\text{limit}} \geq \Delta M_{(i,j',c')} + \sum_{j,c} \Delta M_{(i,j,c)} O_{(i,j,c)\to(i,j',c')} \tag{7}$$

Overall, the optimization target (3) is to minimize the time spent by the longest stage. Constraints (4) and (5) add the sequential dependency requirements on the $F$ and $B$ passes of the same microbatch in adjacent stages. Additionally, (6) adds the dependency constraint imposed by our decision of the scheduling order. Finally, (7) limits the peak activations memory to be below $M_{\text{limit}}$.

Table 8: Comparison between 1F1B, ZB-1p and ZB-2p under the same memory consumption.

| Model | $p$ | $m$ | $b$ | Samples per GPU per second | Memory(GB) | Schedule |
|-------|-----|-----|-----|---------------------------|------------|----------|
| 1.5B | 8 | 24 | 12 | 12.0 | 57 | 1F1B |
| | | | 12 | 13.0 | 61 | ZB-1p |
| | | | 6 | **14.5** | 59 | ZB-2p |
| | | 32 | 12 | 12.6 | 57 | 1F1B |
| | | | 12 | 13.6 | 61 | ZB-1p |
| | | | 6 | **14.8** | 59 | ZB-2p |
| | | 64 | 12 | 13.8 | 57 | 1F1B |
| | | | 12 | 14.4 | 61 | ZB-1p |
| | | | 6 | **14.9** | 59 | ZB-2p |
| 6.2B | 8 | 24 | 6 | 3.56 | 66 | 1F1B |
| | | | 6 | 3.95 | 71 | ZB-1p |
| | | | 3 | **4.32** | 70 | ZB-2p |
| | | 32 | 6 | 3.76 | 66 | 1F1B |
| | | | 6 | 4.05 | 71 | ZB-1p |
| | | | 3 | **4.35** | 70 | ZB-2p |
| | | 64 | 6 | 4.09 | 66 | 1F1B |
| | | | 6 | 4.24 | 71 | ZB-1p |
| | | | 3 | **4.39** | 70 | ZB-2p |
| 14.6B | 16 | 48 | 2 | 1.53 | 50 | 1F1B |
| | | | 2 | 1.73 | 51 | ZB-1p |
| | | | 1 | **1.81** | 51 | ZB-2p |
| | | 32 | 2 | 1.62 | 50 | 1F1B |
| | | | 2 | 1.79 | 51 | ZB-1p |
| | | | 1 | **1.83** | 51 | ZB-2p |
| | | 128 | 2 | 1.78 | 50 | 1F1B |
| | | | 2 | **1.89** | 51 | ZB-1p |
| | | | 1 | 1.85 | 51 | ZB-2p |
| 28.3B | 32 | 96 | 2 | 0.81 | 72 | 1F1B |
| | | | 2 | 0.93 | 74 | ZB-1p |
| | | | 1 | **0.99** | 74 | ZB-2p |
| | | 32 | 2 | 0.85 | 72 | 1F1B |
| | | | 2 | 0.96 | 74 | ZB-1p |
| | | | 1 | **1.00** | 74 | ZB-2p |
| | | 256 | 2 | 0.94 | 72 | 1F1B |
| | | | 2 | **1.02** | 74 | ZB-1p |
| | | | 1 | 1.00 | 74 | ZB-2p |

## H  COMPARE ZB METHODS WITH 1F1B ON SMALL NUMBER OF MICROBATCHES

By nature of PP when the number of microbatches $m$ is less then number of stages $p$, there'll be a large bubble rate. However zerobubble methods can still boost performance under these rare settings by approximately 20% to 30%. In a rough analysis ignoring communication and assuming $m <= p$ and $T_W < T_B$, an 1F1B iteration takes $(m + p - 1) * (T_F + T_B + T_W)$ to complete, while a ZB iteration takes $(m+p-1)*(T_F+T_B)+T_W$. The experiment result is shown in Table 9. Noticeably when $m <= p$ ZB-1p and ZB-2p are essentially the same and consumes similar memory as 1F1B.

Table 9: Comparison between 1F1B and ZB-2p on small number of microbatches.

| Model | $p$ | $m$ | $b$ | Samples per GPU per second | Memory(GB) | Schedule |
|-------|-----|-----|-----|----------------------------|------------|----------|
| 1.5B | 8 | 2 | 6 | 3.56 | 11 | 1F1B |
| | | | 6 | **4.25** | 12 | ZB-2p |
| | | 4 | 6 | 5.74 | 18 | 1F1B |
| | | | 6 | **6.92** | 19 | ZB-2p |
| | | 8 | 6 | 8.26 | 29 | 1F1B |
| | | | 6 | **9.90** | 34 | ZB-2p |
| 6.2B | 8 | 2 | 3 | 1.04 | 21 | 1F1B |
| | | | 3 | **1.33** | 21 | ZB-2p |
| | | 4 | 3 | 1.69 | 28 | 1F1B |
| | | | 3 | **2.16** | 29 | ZB-2p |
| | | 8 | 3 | 2.45 | 39 | 1F1B |
| | | | 3 | **3.07** | 44 | ZB-2p |
| 14.6B | 16 | 4 | 1 | 0.39 | 19 | 1F1B |
| | | | 1 | **0.52** | 20 | ZB-2p |
| | | 8 | 1 | 0.65 | 24 | 1F1B |
| | | | 1 | **0.85** | 24 | ZB-2p |
| | | 16 | 1 | 0.95 | 32 | 1F1B |
| | | | 1 | **1.25** | 33 | ZB-2p |

