# OpenReview forum: "Zero Bubble (Almost) Pipeline Parallelism"
_ICLR.cc/2024/Conference — ICLR 2024 poster_

### Official Review · Reviewer_xbKE · 2023-10-15

**Soundness:** 3 good
**Presentation:** 3 good
**Contribution:** 3 good
**Rating:** 8
**Confidence:** 5

**Summary:**

* The author proposed a new schedule in Pipeline Parallelism (PP) to reduce pipeline bubbles to close to zero
* The key idea is instead of the 1F1B schedule of earlier work, splitting B to two stages (aka activation gradient and weight gradient) and a schedule like 1F1B1W can further reduce bubbles of 1F1B
* The intuition behind it is only Forward calcluation and activation gradient calculation have a pipeline/stage dependency, while weight gradient calculation are not, so activation gradient calcluation should be eagerly scheduled while weight gradient calculation can be rescheduled to fill/balance pipeline bubbles
* The authors have proposed two schedule algorithms to adapt for real-world workloads when F/B/W can have different runtime.

**Strengths:**

* The author made a key observation that the backprop stage in PP can be split into two parts and scheduled in a finer granularity, i.e., `any pipeline (or producer/consumer) independent workload can be rescheduled to balance/fill pipeline bubbles`, which is not only benifitical to train NN with PP, but should hold generally true for any PP workloads

* The author didn't limit themselves to just a simple TF = TB  = TW assumption but also proposed two optimization implementations to optimize the schedule for potentially unbalanced TF/TB/TW

* The authors have conducted representative experiements and showed that their methods can outperform earlier methods by up to 20% and more importantly close to the upperbound (zero bubble) of PP

* The presentation/feagures are easy to follow and illustrative

**Weaknesses:**

* While I agree the authors have discovered a valuable new angle optimizing PP and indeed bring the bubble rate close to zero, I think it is still a bit overselling to claim the method achives zero bubble PP, given:
  1. The method still require a large enough number of microbatches, e.g., m >= 2p to achive an amortized zero bubble PP, how does it perform under m < p is rarely discussed. Experiments (at least a roofline analysis) with m < p can make the methods more sound
  2. There are other methods that can achieve zero bubble PP like PipeDream and PipeMare. These are more strictly flushless (thus bubble-free) methods though they have altered the math of backprop and poses potential accuracy loss or incurs more weight memory. So it is a bit overclaim that this work is the first to achieve pippeline bubbles. Discussions on these previous related works are also encouraged.
  3. Therefore "near zero bubble" is probably more accurate IMHO, emphasis on "zero bubble" actually overshadows the solid intuition mentioned as 1st bullet point in Strengths

* Even though authors proposed two algos to schedule unbalanced TF/TB/TW, the experimetns however are not designed to show how their advantages, e.g., with sentence-size = 1024, TF/TB/TW are almost equal (24h >> 8s in table 1) for all model sizes, experiments or roofline analysis with 8s >> 24h (or any models with sifinicantly different T) could be helpful to evaluate the efficacy of these algos

**Questions:**

1. when authors talk about peak memory I assume they refer to max of peak memory of each worker, rather than the entire pipeline, if it is the latter case even ZB-H1 should have a higher peak memory than 1F1B based on figure 3, can you make this more explicit in the texts?

2. what's a "local search" in 3.1?

3. what's the runtime of ILP, how much does it improve over HEURISTIC? can you do an ablation with HEURISTIC algo only?

4. My understad is section 4 BYPASSING OPTIMIZER SYNCHRONIZATIONS only works for INF/NAN, how does it work for grad-norm when each stage need the global reduced norm? If the optimizer depends on partially reduced state how can it provide same numerical result of baselines like 1F1B?

5. can you try to recompute the activation gradient (not activation), e.g., redo B during W to save some peak memory in H2/2P cases?

---

> ### Author Response · Authors · 2023-11-14
>
> We thank the reviewer for the valuable feedback and suggestions for improvement. We respond to individual points from your review below.
>
> > **Weaknesses:
> Q: The method still require a large enough number of microbatches, e.g., m >= 2p to achive an amortized zero bubble PP, how does it perform under m < p is rarely discussed. Experiments (at least a roofline analysis) with m < p can make the methods more sound**
>
> By the nature of PP, $m < p$ would cause extremely large bubble rate ($\ge 50%$). Previous PP strategies reduce bubble rate by increasing m, for example, in Megatron-LM, all configurations have $m \ge 8p$, while still having a notable bubble rate. Therefore we consider $m < p$ as a very uncommon setting for PP. We want to highlight that compared to previous methods, ZB-2p requires a much smaller number of microbatches to achieve “near zero bubble” (typically $m=3p$ is enough).
>
> > **Weakness:
> Q: There are other methods that can achieve zero bubble PP like PipeDream and PipeMare. These are more strictly flushless (thus bubble-free) methods though they have altered the math of backprop and poses potential accuracy loss or incurs more weight memory. So it is a bit overclaim that this work is the first to achieve pippeline bubbles. Discussions on these previous related works are also encouraged.**
>
> We were aware that asynchronous PP achieves zero bubbles, and intended to briefly touch this point and state that ours is the first to achieve zero bubble under synchronous setting. Somehow we forgot to do so, thanks for pointing this out, we added this discussion in the updated version.
>
> > **Weakness:
> Q: Therefore "near zero bubble" is probably more accurate IMHO, emphasis on "zero bubble" actually overshadows the solid intuition mentioned as 1st bullet point in Strengths**
>
> To avoid over-claiming, we changed the title to “Near Zero Bubble Pipeline Parallelism” in our updated pdf.
>
> > **Weakness:
> Q: Even though authors proposed two algos to schedule unbalanced TF/TB/TW, the experimetns however are not designed to show how their advantages, e.g., with sentence-size = 1024, TF/TB/TW are almost equal (24h >> 8s in table 1) for all model sizes, experiments or roofline analysis with 8s >> 24h (or any models with sifinicantly different T) could be helpful to evaluate the efficacy of these algos**
>
> - We show our profiled value of $T_F/T_B/T_W$ across different settings in Appendix D. From the table, we can see $T_F=T_B$, but $T_W$ is significantly different, with $T_W/T_B$ varies from 0.516 to 0.756.
> - Our algos don’t rely on any assumption on the value of $T_F/T_B/T_W$. The reason why the heuristic works well is because of the flexibility of W and the pattern of “1F-1B-1W”.
> - Actually by dividing W into a finer granularity, e.g. each single weight gradient calculation, it’s easy and straightforward to fill the bubble because the FLOPs of each weight gradient calculation $\ll T_F/T_B$. In practice, we just found that scheduling based on grouping W is already good enough. So we didn’t bring this complexity.

---

> > ### Author Response · Authors · 2023-11-14
> >
> > > **Q1: when authors talk about peak memory I assume they refer to max of peak memory of each worker, rather than the entire pipeline, if it is the latter case even ZB-H1 should have a higher peak memory than 1F1B based on figure 3, can you make this more explicit in the texts?**
> >
> > Yes, the peak memory in paper refers to the maximum value of peak memory of each worker. Thanks for the suggestion, we edited the PDF to make it more clear.
> >
> > > **Q2: what's a "local search" in 3.1?**
> >
> > The “local search” means that we have a hyperparameter to control whether to execute this strategy or not, and we perform a grid search to find the best hyperparameters. We have 3 hyperparameters forming 2^3 combinations, which is a small space, therefore we simply scan through all combinations. The word “local search” is indeed misleading, we rephrased for better clarity in our paper.
> >
> > > **Q3: what's the runtime of ILP, how much does it improve over HEURISTIC? can you do an ablation with HEURISTIC algo only?**
> >
> > As ILP has non-polynomial time complexity, in experiments we limit the runtime of ILP to 2 minutes, initialized with result from heuristic algorithm. For most cases in our experiments ILP shows no significant improvements within the time limit because the heuristic algorithm is already good enough as shown in Table 5. As a result we've also moved the ILP to the Appendix.
> >
> > > **Q4: My understand is section 4 BYPASSING OPTIMIZER SYNCHRONIZATIONS only works for INF/NAN, how does it work for grad-norm when each stage need the global reduced norm? If the optimizer depends on partially reduced state how can it provide same numerical result of baselines like 1F1B?**
> >
> > For grad-norm clipping, the clipping is only triggered when the norm exceeds a certain threshold, otherwise it has no effect. So our strategy is: we greedily update each stage **assuming that clipping will not get triggered**, while at the same time we propagate a partially reduced norm from the first stage to the last stage and finally get the globally reduced norm. If it exceeds the clipping threshold, we inform each stage to rollback and **redo the clipping** like how we deal with NAN. The above strategy is based on the observation that in a stable training gradient clipping is rarely triggered, like INF/NAN also rarely happens.
> > We clarified the process in our updated paper.
> >
> >
> >
> > > **Q5: can you try to recompute the activation gradient (not activation), e.g., redo B during W to save some peak memory in H2/2P cases?**
> >
> > Thanks for the insightful suggestion. We respond in the following aspects.
> > - We would want to highlight ZB-1P/ZB-H1 here. If the 2x activation memory of ZB-2p is a concern, we would prefer ZB-1p over recomputation, because recomputation needs O(#micro-batch) extra computation while ZB-1p only contains O(#pipeline-stage) bubbles.
> > - The peak memory actually happens in the first stage before the first B, where we save a lot of activations for backward pass, so “redo B during W” won’t reduce the peak memory.

---

> > > ### Comment · Reviewer_xbKE · 2023-11-18
> > >
> > > Thanks for all the insightful answers. Regarding Q4, if the model just want to do a grad-norm (e.g., torch.nn.utils.clip_grad_norm_) instead of a conditional grad-clipping, would it still work? since you can't perform a grad-norm clipping (rather than grad clipping) until you get an all-reduced global grad-norm?

---

> > > > ### Author Response · Authors · 2023-11-18
> > > >
> > > > We thank you again for your positive feedback and encouraging support for raising the score. For the Q4, we use the property that most of the time gradient clipping does not require global information because the clip ratio is exactly 1 when global_grad_norm $\le$ max_norm, and only rollback when global_grad_norm $>$ max_norm. However, when global information is always needed (always normalize gradients by max_norm/global_grad_norm), synchronization is inevitable.
> > > >
> > > > However, we notice that the example of torch.nn.utils.clip_grad_norm_ you mentioned also aligns with our use case. From the [source code](https://pytorch.org/docs/stable/_modules/torch/nn/utils/clip_grad.html#clip_grad_norm_), it also rescales the gradients by max_norm/global_grad_norm (See the code below) and clip the ratio to maximum 1. When the clip ratio is clipped to exactly 1, no scaling on gradients will be performed.
> > > > ```
> > > > clip_coef = max_norm / (total_norm + 1e-6)
> > > > # Note: multiplying by the clamped coef is redundant when the coef is clamped to 1, but doing so
> > > > # avoids a `if clip_coef < 1:` conditional which can require a CPU <=> device synchronization
> > > > # when the gradients do not reside in CPU memory.
> > > > clip_coef_clamped = torch.clamp(clip_coef, max=1.0)
> > > > for ((device, _), ([grads], _)) in grouped_grads.items():  # type: ignore[assignment]
> > > >     if (foreach is None or foreach) and _has_foreach_support(grads, device=device):  # type: ignore[arg-type]
> > > >         torch._foreach_mul_(grads, clip_coef_clamped.to(device))  # type: ignore[call-overload]
> > > >     elif foreach:
> > > >         raise RuntimeError(f'foreach=True was passed, but can\'t use the foreach API on {device.type} tensors')
> > > >     else:
> > > >         clip_coef_clamped_device = clip_coef_clamped.to(device)
> > > >         for g in grads:
> > > >             g.detach().mul_(clip_coef_clamped_device)
> > > > ```

---

> > > > > ### Comment · Reviewer_xbKE · 2023-11-18
> > > > >
> > > > > Okay, somehow I forgot when clip_coef < 1, grad is not scaled. Good point, thanks.

---

> > ### Comment · Reviewer_xbKE · 2023-11-18
> >
> > Thanks for the edits and additional experiments. I am happy to revise my score to 8 and look forward to the acceptance of your work.

---

> ### Author Response · Authors · 2023-11-16
>
> Additional info for:
> > **Weaknesses:
> Q: The method still require a large enough number of microbatches, e.g., m >= 2p to achive an amortized zero bubble PP, how does it perform under m < p is rarely discussed. Experiments (at least a roofline analysis) with m < p can make the methods more sound**
>
> We compared ZB methods when $m\le p$ and added the result to appendix. Roughly the run time of 1F1B and ZB methods when $m\le p$ are $(m+p-1)(T_F+T_B+T_W)$  vs  $(m+p-1)(T_F+T_B)+T_W$. The experiment also shows that we can still get approximately 20%~30% improvement in these settings.

---

### Official Review · Reviewer_b5Up · 2023-10-25

**Soundness:** 2 fair
**Presentation:** 2 fair
**Contribution:** 2 fair
**Rating:** 6
**Confidence:** 4

**Summary:**

Pipeline parallelism is a widely used technique in distributed training. However, the efficiency of pipeline parallelism suffers from pipeline bubbles. In view of this, this paper designs a new strategy to reduce the bubble rates. The idea behind the new pipeline strategy is to split the backward computation into two parts, which could reduce the bubble in the 1F1B strategy.

**Strengths:**

1. The problem studied in this paper is very important. With the increase of the model parameters, pipeline parallelism is a popular strategy for training large models. However, the bubble in the 1F1B strategy affects the efficiency of the pipeline strategy.
2. The idea behind this paper is very easy. It is a good signal, I think the paper develops an easy solution with great performance.
3. This paper contains a mathematical analysis.

**Weaknesses:**

1. The authors miss an important point in the experiment section.
2. The writing of this paper can be slightly improved.
3. The performance of the method developed in this paper is limited.

**Questions:**

The problem studied in this paper is significant in distributed training and the authors give an excellent solution to improve the existing pipeline strategy.

However, the main issue of this paper is that the solution mentioned in this paper is likely to increase the memory usage of pipeline parallelism training. Moreover, the experimental results introduce bubble rate which is not important in the pipeline parallelism training. It is known to all that the efficiency of pipeline parallelism suffers from pipeline bubbles. But the bubble rate is not important. We only care about the throughput and memory usage of the pipeline parallelism strategy. Since the pipeline strategy developed in this paper will increase the memory usage of pipeline parallelism training, large memory usage is likely to decrease the size of the micro-batch. A small micro batch will decrease the efficiency of the pipeline parallelism strategy. The experimental results show the developed algorithm has limited improvement.

The writing of this paper could be improved. For example, we can use different colors to represent B in Figure 3 and Backward in Figure 2. Moreover, the title of this paper is too large. I do not think the method in this paper is zero bubble.

---

> ### Author Response · Authors · 2023-11-14
>
> We thank the reviewer for the valuable feedback. We respond to individual points from your review below.
>
> > **Q1 & Q2. … likely to increase the memory usage of pipeline parallelism training, … care about memory usage**
>
> We want to highlight the automatic scheduling algorithm which provides the flexibility for users to trade-off between memory usage and bubble rate by themselves. Both ZB-1p and ZB-2p are special cases generated from this algorithm with different memory limits. Although ZB-2p indeed needs a doubled memory, ZB-1p still significantly outperforms 1F1B under a similar memory constraint.
>
>
>
> > **Q2. bubble rate is not important. We only care about the throughput and memory usage.**
>
> As throughput $\propto$ (1 - bubble rate) * (kernel efficiency), where bubble rate measures how good the pipeline schedule is, and kernel efficiency measures how a single op is utilizing the GPU. We report bubble rate separately to reflect how efficient our pipeline schedule is, as this is the main contribution of this work. The comparison is under the same batch size and thus the same kernel efficiency.
>
> We also presented throughput in Table 3.
>
> > **Q3. A small micro batch will decrease the efficiency**
>
> We double the size of each microbatch for 1F1B and compare its throughput with ZB-2p. The experimental results show that ZB-2p also holds a higher throughput than 1F1B by about 14% on average (min 4%, max 22%), even with a half size for each microbatch. Empirically, a larger batch size increases the utilization rate of the GPU and thus improves the efficiency. However, it is less of a concern for large models because the hidden dimension is big enough to saturate device utilization. Based on this consideration and our experimental results, we believe ZB-2p is preferred over increasing the batch size for 1F1B in LLM training. For more details of the experiments, please see the Appendix F in our updated pdf.
>
> > **Q4. The experimental results show the developed algorithm has limited improvement.**
>
> Despite the memory concern, in an apple to apple comparison where our memory usage is similar to the baselines, our method ZB-1p improves the throughput by 9.5% on average (min 4%, max 15%, row 2 and 3 in table 4). With larger models (14.6B & 28.3B), the average throughput improvement is 11.7%. We think these numbers are already significant considering the cost of GPU. Furthermore, when memory allows, our method pushes the throughput almost to the theoretical upper limit (red bar in figure 5 and bubble rates of ZB-2p in Table 5).
>
> Another advantage of ZB-2p is that it can achieve near zero bubble with a smaller number of micro-batch (typically m=3p is enough), for 1F1B in Megatron the setting is m>=8p yet still has notable bubbles. This means more micro-batch can be partitioned over Data Parallelism (DP) dimension which brings a better scalability for the training of LLM. Given a fixed PP size and a fixed global batch size, we can reduce #micro-batch of PP to a relatively small number (e.g. 3p) and increase the size of DP, thus shortening the training time by using more devices. When integrated with ZeRO, we can further partition the parameter memory (including optimizer states) to compensate for the extra activation memory introduced by ZB-2p.
>
>
> > **Q5. use different colors to represent B in Figure 3 and Backward in Figure 2.**
>
> We adjusted accordingly in the updated PDF.
>
> > **Q6. title of this paper is too large**
>
> It is true that the bubble rate can never go to zero because practically there’s always a difference in computation time of F/B/W, introducing gaps in between. As Reviewer xbKE also suggested “Near Zero Bubble” could be more accurate and also well supported by our experimental evidence, we updated the title accordingly in the PDF.

---

> > ### Comment · Reviewer_b5Up · 2023-11-22
> >
> > Thanks for your reply! I am happy to raise my score to 6.

---

> > > ### Author Response · Authors · 2023-11-22
> > >
> > > We truly appreciate your time in revisiting our response and the positive feedback, many thanks!

---

### Official Review · Reviewer_Uec3 · 2023-10-31

**Soundness:** 2 fair
**Presentation:** 3 good
**Contribution:** 2 fair
**Rating:** 6
**Confidence:** 3

**Summary:**

The manuscript titled "ZERO BUBBLE PIPELINE PARALLELISM" presents a novel strategy for achieving pipeline parallelism with the primary goal of eliminating pipeline bubbles. This approach includes the development of an automatic scheduling algorithm, which is then compared with established methods like 1F1B and interleaved 1F1B. The key contributions of the paper are as follows:

1. It introduces a pipeline parallelism strategy designed to eliminate pipeline bubbles, assuming that forward, backward, and weight gradient calculations all take the same amount of time.

2. It presents an automatic scheduling algorithm that consistently outperforms traditional methods, with the highlight being the achievement of a zero bubble rate, indicating optimal computational resource utilization. However, this comes at the cost of nearly doubling memory usage.

3. The paper delves into the relationship between memory constraints and bubble rates, providing insights into the trade-offs between memory requirements and scheduling efficiency.

**Strengths:**

The paper introduces a novel strategy for pipeline parallelism aimed at completely eliminating pipeline bubbles. Leveraging this strategy, an automatic scheduling algorithm is developed and benchmarked against existing approaches, such as 1F1B and interleaved 1F1B.

originality: fair. The paper separated the backward loss and weight gradient calculation, and achieved zero bubble pipeline parallelism under the assumption that forward, backward loss, and weight gradient calculation time are identical.
quality: good. The paper has theoretical calculation and also experimental data to support the zero bubble pipeline parallelism algorithm proposed in this paper.
clarity: good.
significance: medium. The algorithm can achieve zero bubble pipeline parallelism but with memory usage nearly doubled. This will limit the application of this method for larger LLM given larger models will use larger memory.

**Weaknesses:**

The algorithm can achieve zero bubble pipeline parallelism but with memory usage nearly doubled. This will limit the application of this method for larger LLM given larger models will use larger memory.

**Questions:**

For Figure 5, sometimes ZB-1p is better than 1F1B-I, and sometime not. Can you explain the reason in the paper?

---

> ### Author Response · Authors · 2023-11-14
>
> We thank the reviewer for the valuable feedback. We respond to individual points from your review below.
>
> > **Weakness: memory usage limit application to LLM**
>
> For the concern of application to LLM, we want to highlight the following points.
> * We want to highlight the automatic scheduling algorithm which provides the flexibility for users to trade-off between memory usage and bubble rate by themselves. Both ZB-1p and ZB-2p are special cases generated from this algorithm with different memory limits. Although ZB-2p indeed needs a doubled memory, ZB-1p still significantly outperforms 1F1B under a similar memory constraint.
> * We double the size of each microbatch for 1F1B and compare its throughput with ZB-2p. The experimental results show that ZB-2p also holds a better performance even with a half size for each microbatch. Empirically, a larger batch size increases the utilization rate of the GPU and thus improves the efficiency. However, it is less of a concern for large models because the hidden dimension is big enough to saturate device utilization. Based on this consideration and our experimental results, we believe ZB-2p is preferred over increasing the batch size for 1F1B in LLM training. For more details of the experiments, please see the Appendix F in our updated pdf.
> * Another advantage of ZB-2p is that it can achieve near zero bubble with a smaller number of micro-batch (typically m=3p is enough), for 1F1B in Megatron the setting is m>=8p yet still has notable bubbles. This means in ZB-2p, more micro-batch can be partitioned over Data Parallelism (DP) dimension which brings a better scalability for the training of LLM. Given a fixed PP size and a fixed global batch size, we can reduce #micro-batch of PP to a relatively small number (e.g. 3p) and increase the size of DP, thus shortening the training time by using more devices. When integrated with ZeRO, we can further partition the parameter memory (including optimizer states) to compensate for the extra activation memory introduced by ZB-2p.
>
>
> > **Question: For Figure 5, sometimes ZB-1p is better than 1F1B-I, and sometime not. Can you explain the reason in the paper?**
>
> From Table 5, we can find that 1F1B-I has lower bubble rates for models of 1.5B, 6.2B and 28.3B, and a bit higher bubble rate for the 28.3B model. However, 1F1B-I is generally under-performed than theoretical efficiency in our experiments, especially in multi-node setups. We guess the reason is that 1F1B-I has more communication and computation passes than both 1F1B and ZB-1p, so it is more likely to be affected by the fluctuations in the time cost of each pass. For multi-node setups, the time cost of inter-node communication is usually larger than intra-node communication, which also decreases the implemental efficiency of 1F1B-I because we don’t take this difference into account.

---

### Official Review · Reviewer_hNFq · 2023-11-01

**Soundness:** 3 good
**Presentation:** 3 good
**Contribution:** 4 excellent
**Rating:** 8
**Confidence:** 4

**Summary:**

The paper presents a novel approach, called ZeroBubble, for eliminating bubbles in pipeline parallelism in order to training efficiency. There are two key aspects to the proposal: (1) splitting the backward pass into the B and W components to increase scheduling flexibility, and (2) eliminating synchronization of optimizer step with a rollback mechanism to preserve semantics. The paper also presents various analysis to illustrate the memory consumption effects of different ZeroBubble strategies. Overall, the evaluation results show that ZeroBubble provides up 30% throughput improvement over 1F1B schedules for Megatron GPT models.

**Strengths:**

I think splitting the backward pass into B (activation gradient computation) and W (weight gradient computation) as a way of improving scheduling flexibility is a nice touch of creativity.

The paper includes ample analytical details that foster intuition and understanding of the relevant memory and TFLOPs consideration of pipeline parallelism schedules.

The authors did a really great job of writing and organizing the paper.

**Weaknesses:**

My main concern relates to breaking the synchronization of optimizer step because it complicates the synchronous training semantics, and adoption requires close interaction of rollback and model checkpointing.  Moreover, I think there are few design alternatives that the paper failed to explore or discuss.
1. Evaluating the performance of synchronous optimizer step as way of understanding the trade-off between the simplicity and throughput.
2. Scheduling W before B, which is less memory-efficient but allows staggering the optimizer step of the stages while preserving the synchronous semantics.

Another concern, or perhaps a question is whether the more memory efficient schemes (1F1B* and ZB-1p) be evaluated with larger micro-batch sizes than ZB-2p. This would help to confirm that the evaluation is not biased towards ZB-2p.

**Questions:**

See weakness.

---

> ### Author Response · Authors · 2023-11-14
>
> We thank the reviewer for the positive feedback and suggestions for improvement. We respond to individual points from your review below.
>
> >**adoption requires close interaction of rollback and model checkpointing**
>
> We guess the “model checkpointing” you mentioned here means saving previous model parameters/optimizer states and restoring them when rollback.
> - For efficiency, we recommend the in-place optimizer rollback proposed in Appendix C, which does not require checkpointing or extra copy of parameters and should be more efficient.
> - For simplicity, from our study, gradient clipping and NAN are mainly activated during the initial stage of training, which we believe is more a choice for training stability than model performance. Therefore, instead of post validation, there’re indeed other simpler choices that may achieve the same. For example, clipping the gradients locally. Since in this work we would like to focus on the systems aspect, we did not explore the numerical/algorithmic choices, but just align it with our baselines.
>
> > **Weakness 1:
> > Evaluating the performance of synchronous optimizer step as way of understanding the trade-off between the simplicity and > throughput.**
>
> Thanks for the valuable suggestion. We add some new experiments in Appendix E, which shows that the synchronous optimizer under-performs our post-validation strategy by about 8.8% on average.
>
> > **Weakness 2:
> > Scheduling W before B, which is less memory-efficient but allows staggering the optimizer step of the stages while preserving the synchronous semantics.**
>
> Thanks for the suggestion, we interpret your point as that if W goes earlier than B, then the optimizer step can start earlier and overlap with B, is it correct? However, for a neural network W in layer i is dependent on B in layer i+1, this dependency prevents us from shifting W to the front enough to achieve the described benefit.
>
>
> > **Another concern, or perhaps a question is whether the more memory efficient schemes (1F1B\* and ZB-1p) be evaluated with larger micro-batch sizes than ZB-2p. This would help to confirm that the evaluation is not biased towards ZB-2p.**
>
> We add experiments comparing ZB-2p and 1F1B (with double micro-batch size) under the same memory consumption. The results show that ZB-2p also has a higher throughput than 1F1B by about 14% on average (min 4%, max 22%). Please find related data in Appendix F in the updated manuscript.

---

### Author Response · Authors · 2023-11-14

Thanks for all reviewers for the valuable feedbacks and insightful suggestions. We updated our PDF accordingly, with changes marked as red color. The changes mainly include:
- Add experiments in Appendix F, increasing the microbatch size for 1F1B/ZB-1p and compare their thoughput with ZB-2p. The results show that ZB-2p also obviously outperforms than 1F1B by about 14% on average (min 4%, max 22%), even under the same memory consumption.
- Add ablation study of optimizer post-validation strategy in Appendix E, which shows it improves the throughput by about 8.8% compared to all-reduce synchronization.
- Change the tile of paper to "Near Zero Bubble Pipeline Parallelism" to avoid over-claiming.
- Emphasizing it is under sychronous training semantic where we achieve zero bubble in abstract, which is accidentally missed in previous version.
- Clarify some unclear points in heuristic algorithm and optimizer post-validation strategy.
- Add experiments in Appendix H for cases where number of microbatches is less or equal than number of pipelines.
- Move ILP formulation to Appendix.
- Reconstruct conclusion part.

---

### Author Response · Authors · 2023-11-17

Dear Reviewers,

Thank you again for your valuable comments and suggestions, which are really helpful for us. We have posted responses to the proposed concerns.

We totally understand that this is quite a busy period, so we deeply appreciate it if you could take some time to return further feedback on whether our responses solve your concerns. If there are any other comments, we will try our best to address them.


Best,

The Authors

---

### Author Response · Authors · 2023-11-21

Dear Reviewers,

We would like to express our sincere gratitude for your insightful comments and suggestions, which are immensely valuable to us. We have provided detailed responses addressing the raised concerns.

Understanding the time constraints you face, we highly appreciate your consideration in revisiting our responses to ensure they adequately address your concerns. Should there be any additional comments, we are committed to making every effort to address them.

Best regards,

The Authors

---

### Meta-Review · Area_Chair_1SUj · 2023-12-05

**Metareview:**

The paper proposed a new scheduling strategy for distributed training. The scheduler introduces nearly no pipeline bubbles (idle time). The proposed approach is about 10% more efficient than prior approaches and close to the global optimum (upper bound).
The reviewers remarked
+ The paper is well written and easy to read.
+ The problem studied is interesting
+ The solution is novel and performs well
Most concerns have been addressed in the rebuttal.
Overall all reviewers recommend acceptance. The AC agrees.

**Justification For Why Not Higher Score:**

The paper is well written and executed, but of fairly narrow interest. I'm not opposed to spotlight, but poster seems to be a better fit.

**Justification For Why Not Lower Score:**

Should be accepted.

---

### Decision · Program_Chairs · 2024-01-16

Accept (poster)